# Self-Powered, Non-Toxic, Recyclable Thermogalvanic Hydrogel Sensor for Temperature Monitoring of Edibles

**DOI:** 10.3390/mi14071327

**Published:** 2023-06-28

**Authors:** Kun Yang, Chenhui Bai, Boyuan Liu, Zhoutong Liu, Xiaojing Cui

**Affiliations:** 1College of Electronic Information and Optical Engineering, Taiyuan University of Technology, Taiyuan 030024, China; 13111189389@163.com (C.B.); lby151831868@163.com (B.L.); liuzhoutong2023@163.com (Z.L.); 2Shanxi Transportation Technology Research & Development Co., Ltd., Taiyuan 030032, China; 3School of Physics and Information Engineering, Shanxi Normal University, Taiyuan 030031, China

**Keywords:** hydrogel, thermogalvanic, gelatin, self-powered

## Abstract

Thermogalvanic hydrogel, an environmentally friendly power source, enable the conversion of low-grade thermal energy to electrical energy and powers microelectronic devices in a variety of scenarios without the need for additional batteries. Its toxicity, mechanical fragility and low output performance are a hindrance to its wide application. Here, we demonstrate thermoelectric gels with safe non-toxic, recyclable, highly transparent and flexible stretchable properties by introducing gelatin as a polymer network and SO${{}_{3/4}}^{2-}$ as a redox electric pair. When the temperature difference is 10 K, the gel-based thermogalvanic cell achieves an open-circuit voltage of about 16.2 mV with a maximum short-circuit current of 39 μA. Furthermore, we extended the application of the Gel-SO${{}_{3/4}}^{2-}$ gel to monitor the temperature of hot or cold food, enabling self-powered sensing for food temperature detection. This research provides a novel concept for harvesting low-grade thermal energy and achieving safe and harmless self-driven temperature monitoring.

## 1. Introduction

The burgeoning growth of the wearable electronics market has ushered in a profound transformation, compelling a remarkable surge in demand for self-sufficient energy sources to sustain the burgeoning array of devices [1,2,3]. This dynamic evolution has, in turn, catalyzed a significant upsurge in research and development endeavors focusing on the extraction and utilization of low-grade waste heat. Such an escalating interest stems from the realization that harnessing and effectively converting this abundant but often underutilized energy source can address the pressing need for sustainable and eco-friendly power solutions in the realm of wearable electronics. As a result, the quest for innovative techniques and materials for efficient waste heat collection has become an imperative in the academic discourse surrounding this ever-evolving field [4,5,6]. The human body maintains an approximate surface temperature of 306 K, resulting in a temperature difference of approximately 7 K compared to the surrounding environment [7,8,9]. There is also a temperature difference of more than 10 K or even 20 K between food and the environment. Typically, a significant portion of this low-grade heat is wasted, contributing to various environmental issues [10,11,12,13]. Moreover, with the advancements in wearable electronics and e-skins, there is a growing recognition of the necessity for self-sustaining energy supplies, leading to increased attention to energy harvesting from humans or the environment [14,15].

However, traditional thermoelectric generators primarily rely on solid-state semiconductors or conductive polymers, but their output voltage is restricted due to their low Seebeck coefficient, typically only a few hundred μV·K${}^{-1}$ [16,17]. Moreover, these conventional thermoelectric materials often lack the desired flexibility and stretchability required for conformity in practical applications [18,19,20]. As a result, thermogalvanic hydrogel has offered a wide array of potential applications in the field of body energy harvesting and wearable electronics [21,22,23,24]. Thermogalvanic hydrogel is electrochemical cells with the introduction of redox pairs in hydrogel systems, which experienced a renaissance in the last decade with respect to harvesting low-grade thermal energy [9,25,26].

Nevertheless, it is important to note that certain thermogalvanic materials, such as Fe${}^{2+/3+}$, Fe(CN)${}_{6}^{3-/4-}$, Co(bpy)${}_{3}$Cl${}_{2/3}$ [27], may contain toxic metal elements, which makes them unsuitable for direct skin contact. The potential toxicity associated with these materials raises concerns regarding their practical application, particularly in wearable electronics and medical devices [15]. Therefore, there is a growing need to explore alternative thermogalvanic materials that are non-toxic, environmentally friendly and safe for use in contact with the skin. To address this issue, recent research has focused on the development of thermocouple ion-based aqueous thermocells, which offer a promising solution for achieving safe skin thermal energy conversion [28]. These innovative thermocells utilize ion-based aqueous electrolytes instead of toxic metal-containing materials, ensuring a non-toxic and environmentally friendly approach to energy harvesting. By exploiting the unique properties of these ion-based electrolytes, researchers aim to achieve efficient thermoelectric conversion while prioritizing safety and compatibility with the human body. Moreover, there is an increasing interest in the development of wearable medical electronics with improved thermoelectric conversion efficiency. These devices require materials that are not only efficient in converting thermal energy into electrical energy but also meet stringent safety requirements. The pursuit of non-toxic, environmentally friendly and recyclable thermoelectric materials has gained significant attention in recent years. Such materials would not only enable efficient energy harvesting from the human body or the environment but also contribute to sustainable and eco-friendly wearable electronic technologies. To ensure the safe conversion of thermal energy from the skin, researchers have devoted significant efforts to exploring novel approaches, one of which includes the development of novel thermocouple ion-based aqueous thermocells. These innovative thermocells represent a promising solution for achieving efficient and safe energy conversion [26,29]. In traditional thermoelectric generators, solid-state semiconductors or conductive polymers have been widely used. However, their limited output voltage, primarily due to their low Seebeck coefficient of only a few hundred μV·K${}^{-1}$, poses a challenge for practical applications. Moreover, these traditional thermoelectric materials often lack flexibility and stretchability, making it difficult for them to conformably adhere to the human body [30]. Consequently, there is a surging interest in the development of wearable medical electronics that exhibit high thermoelectric conversion efficiency, while being non-toxic, environmentally friendly and recyclable. Sodium sulfite (Na${}_{2}$SO${}_{3}$) and sodium sulfate (Na${}_{2}$SO${}_{4}$) are widely recognized as common inorganic salt compounds. These substances are generally regarded as relatively non-toxic under normal conditions. They have been extensively used in various fields, including medicine, the food industry and environmental science, where their safety and non-toxic nature have been well-established [31]. Considering their favorable properties and wide applicability, sodium sulfite and sodium sulfate serve as suitable choices as redox pairs for thermoelectric gels. Their established safety profiles provide reassurance regarding the use of these compounds in thermogalvanic gel systems. By selecting sodium sulfite (Na${}_{2}$SO${}_{3}$) and sodium sulfate (Na${}_{2}$SO${}_{4}$) as redox pairs, the thermoelectric gels benefit from the safe and reliable nature of these compounds [32]. This choice ensures that the resulting thermogalvanic gel materials can be used with confidence in various applications, including wearable electronic devices and temperature-sensing systems. Furthermore, the utilization of sodium sulfite and sodium sulfate in thermogalvanic gels opens up possibilities for further exploration and advancements in this field [33,34,35]. Researchers can continue to investigate their performance, optimize their properties and explore novel applications to harness the potential of these safe and reliable redox pairs for thermoelectric energy conversion and temperature monitoring [36,37].

Herein, a gelatin-based thermogalvanic hydrogel with non-toxic redox couple of SO${}_{3/4}^{2-}$ as a redox pair is proposed. The one-pot method is used in the preparation of the Gel-SO${}_{3/4}^{2-}$ thermoelectric gel. The gelatin monomers with the characteristics of being safe and edible are used as a polymer network to construct adaptable thermoelectric systems. Compared with conventional thermoelectric gels, this thermoelectric gel not only has excellent thermoelectric and flexible stretchable properties but also offers excellent safety and multiple recyclabilities to ensure sufficient long-term use. Based on the possibility of global sustainability, the proposed thermoelectric gels can provide a future outlook for smart wearable electronics.

## 2. Experimental Section

### 2.1. Materials

The raw materials are as follows: gelatin (photographic grade, adhesive strength ∼250 g), NaCl (MW = 58.44, spectral level), Na${}_{2}$SO${}_{4}$ (MW = 142.04, 99%), Na${}_{2}$SO${}_{3}$ (MW = 126.04, 99%). The above materials were purchased from Aladdin Industrial Corporation. Deionized water (resistivity of 18.2 MΩ·cm) in all experiments was prepared via a Milli-Q Direct 8 ultrapure water purification system. All chemical reagents were employed without further purification.

### 2.2. Preparation of the Gel-SO${}_{3/4}^{2-}$ Gel

Firstly, gelatin with a mass concentration of 30% was dissolved in a stirring setup at a temperature of 65 °C. This dissolution process took place over a period of 3 h, resulting in the formation of solution 1. Next, a mixture of sulfate and sulfite in a stoichiometric ratio of 1:1 was prepared with a concentration of 0.2 mol/L. This mixture, referred to as solution 2, was thoroughly stirred to ensure homogeneity. To complete the gel formulation, NaCl was added to solution 2, achieving a concentration of 0.6 mol/L. The resulting mixture, denoted solution 3, underwent thorough stirring to ensure proper mixing of the components. The next step involved the combination of solutions 1 and 3. These two solutions were mixed together and stirred at a temperature of 65 °C for a duration of 30 min. This stirring process facilitated the uniform distribution and integration of the gelatin and ionic components, resulting in the formation of a homogenous gel solution. Finally, the gel-filled molds were allowed to cool down, leading to the solidification of the gel structure. The cooling process enabled the gel solution to transition into a stable gel state, forming the desired gelatin-based thermogalvanic gel.

### 2.3. Recasting of Gel-SO${}_{3/4}^{2-}$ Gel

Firstly, the gel is carefully cut or chopped into small pieces with a diameter of less than 5 mm. This size reduction facilitates the subsequent melting process and promotes uniform heating throughout the gel material. Next, the gel pieces are subjected to heating at a temperature of 70 °C. The application of heat causes the gel pieces to melt and transform into a transparent liquid. This liquefied state allows for easy handling and manipulation of the gel material. Finally, the liquefied gel is injected into a fixed touching apparatus or mold, where it is shaped according to the desired form. The apparatus or mold is designed to provide the gel with the intended shape and dimensions. Upon cooling, the gel undergoes solidification, resulting in the desired shape of the thermogalvanic gel. This recovery process enables the efficient reuse and reshaping of the Gel-SO${}_{3/4}^{2-}$ gel. By following these steps, the gel material can be regenerated and transformed into its original form, ensuring its continued usability and versatility for future applications.

### 2.4. Tensile/Compression Test

The tensile and compression properties of the hydrogels were evaluated using a Universal Tensile Testing machine (UTM, Qualitest, Chicago, IL, USA) at room temperature (*T* = 293 K) and relative humidity of 50%. For the tensile tests and cyclic tests, the hydrogel materials were prepared as strips with dimensions of 30 × 10 × 3 mm. The testing machine applied a loading speed of 200 mm·min${}^{-1}$ to apply tensile force to the samples. In addition to the tensile tests, compression tests were also conducted on the hydrogel samples. The samples were shaped into columns with a diameter of 15 mm and a height of 25 mm to perform the compression tests and cyclic tests. The testing machine applied a loading speed of 50 mm·min${}^{-1}$ to compress the samples. During the testing process, Young’s modulus values, which represent the stiffness of the materials, were directly obtained using the software integrated with the UTM. To ensure the reliability of the obtained results, each Gel-SO${}_{3/4}^{2-}$ gel sample underwent testing with three replicates. This approach helped to account for any variations or inconsistencies that may arise during the testing process and ensured the accuracy and validity of the measured tensile and compression properties.

### 2.5. Ionic Conductivity

The electrical conductivity of the hydrogel was characterized using an electrochemical workstation (CHI660E, China) in combination with cyclic voltammetry (CV) measurements. The CV measurements were performed by sweeping the voltage from −1 V to +1 V. This allowed us to analyze the electrical response of the Gel-SO${}_{3/4}^{2-}$ gel over a range of applied voltages. To determine the ionic conductivity of the Gel-SO${}_{3/4}^{2-}$ gel, impedance spectroscopy was employed. The impedance measurements were conducted in a frequency range of 10${}^{-1}$–10${}^{6}$ Hz. During the measurements, a block-shaped hydrogel sample with dimensions of 20 × 20 × 10 mm was prepared. To facilitate the measurements, a piece of Gel-SO${}_{3/4}^{2-}$ gel was placed between two sheets of carbon paper, which served as the electrode materials. The ionic conductivity (σ) of the Gel-SO${}_{3/4}^{2-}$ gel was calculated using the following formula: σ=d/(R×S), where the area of the hydrogel (*S*) is 1 cm${}^{2}$, the distance between the electrodes (*d*) represents the thickness of the hydrogel sample and the volume resistance (*R*) is determined by the intercept with the real axis on the impedance spectrum.

## 3. Results and Discussion

Thermogalvanic hydrogels have shown promise in utilizing liquid electrolytes containing redox couples such as Fe${}^{2+/3+}$, Fe(CN)${}_{6}^{3-/4-}$ and Co(bpy)${}_{3}$Cl${}_{2/3}$, which exhibit a relatively high thermoelectric power of several mV·K${}^{-1}$ in the presence of a temperature gradient [38]. Figure 1a illustrates a schematic diagram of the exemplary gel. Here, the redox couple of SO${}_{3/4}^{2-}$ is chosen as the thermogalvanic ions because it is free of heavy metal ions that are harmful to humans.

A non-toxic, self-healing and recyclable thermoelectric gel is successfully developed by utilizing gelatin as a polymer network structure, the preparation process of the gel is shown in Appendix A. The implanted redox couple in the gel enables seamless transfer between the two terminals without any hindrance. SEM images in Appendix A reveal the porous cross-linked morphology of the Gel-SO${}_{3/4}^{2-}$ gel. It is worth noting that other redox couples and ions that can promote a scalable network and efficient dispersion within the matrix are also suitable for the approach. For a thermogalvanic hydrogel with SO${}_{3/4}^{2-}$, a reversible redox reaction SO${}_{3}^{2-}$↔ SO${}_{4}^{2-}$ + e persists in a temperature gradient (Figure 1b). On the cathode electrode, the SO${}_{3}^{2-}$→ SO${}_{4}^{2-}$ + e oxidation reaction occurs due to the favorable electrochemical reaction and electrons are induced from the cathode electrode, leading to a lower electrode potential and a rise in electrochemical potential. This continuous cyclic reaction between the anode and cathode electrodes occurs due to the thermodynamically favorable reduction reaction of SO${}_{4}^{2-}$ to SO${}_{3}^{2-}$ on the hot electrode. This causes high temperature end emission of electrons, leading to an increase in electrochemical potential and a decrease in electrode potential. Through convection, diffusion and migration, the oxidized SO${}_{4}^{2-}$ ions are transported to the low-temperature electrode, while the reduced SO${}_{3}^{2-}$ ions travel back to the low-temperature electrode. As a result, voltages are generated between the two electrodes in a quasi-continuous manner under the temperature gradient. As shown in Figure 1c, in order to investigate the temperature gradient and the corresponding potential building process of the Gel-SO${}_{3/4}^{2-}$ gel, the finite element simulation is employed to analyze the passive heat transfer behavior of the Gel-SO${}_{3/4}^{2-}$ gel as a thermally conductive network. It is observed that as the heating time increases, the Gel-SO${}_{3/4}^{2-}$ gel can establish a fast and uniform temperature gradient, which leads to the establishment of a thermoelectric potential.

To further explore the mechanical property of the Gel-SO${}_{3/4}^{2-}$ gel, a series of strain–stress characterizations are conducted. During the tensile tests, the Gel-SO${}_{3/4}^{2-}$ thermogalvanic gel shows high stretchability and excellent toughness, which are illustrated in Figure 2a. The un-notched Gel-SO${}_{3/4}^{2-}$ gel can be stretched to a strain of 350%, with a high Young’s modulus of 0.22 MPa and a high strength of 0.69 MPa. Even for 20% and 50% notched specimens, it can be stretched to about 180% and 150% strain, respectively, which can indicate that the energy concentrated at the notch can be dissipated during the stretching process. The tensile test in Figure 2b shows that A-gel has favorable tensile strength and excellent transparency. The results of the tensile cycling test are presented in Figure 2c–e, where the fatigue resistance of the organogel is assessed through 10 successive loading–unloading tests at deformation levels of 50%, 70% and 100%. The loading–unloading profiles of the Gel-SO${}_{3/4}^{2-}$ gel at different strains exhibit noticeable hysteresis loops, which become more pronounced with increasing strain. The maximum stress gradually increases, likely due to the recombination of hydrogen bonds. Notably, the dissipation energy value of the gel in the first loading cycle is significantly higher than in subsequent cycles, indicating substantial softening after the initial loading. However, in the subsequent 2nd–10th loading–unloading cycles, the hysteresis loops almost overlap with the second cycle (Figure 2e), indicating the organogel has a rapid self-recovery capability and remarkable fatigue resistance. Additionally, the optimized hydrogels demonstrate favorable compressive performance and stable compressive cycles (Figure 2f). These findings highlight the excellent mechanical properties and fatigue resistance of the gel, making it well-suited for practical applications in wearable electronics and beyond.

The thermal conductivities of the gels depicted in Figure 3a were measured via the steady-state method, which gives the values of about 0.31 and 0.26 W· m${}^{-1}$K${}^{-1}$ for the gelatin hydrogels and Gel-SO${}_{3/4}^{2-}$ gel at the temperature of 283K, respectively. This indicates that a stable temperature gradient can be established. Figure 3b illustrates the relationship between ionic conductivity and frequency. It can be observed that with the increase in gel temperature, the ionic conductivity exhibits a corresponding upward trend and the conductivity also increases as the frequency rises. Typically, the gel conductivity is measured at a frequency of 0.1 Hz since the thermoelectric gel generates direct current. This frequency is considered suitable for accurately evaluating the gel’s conductivity performance. Figure 3c shows the electrochemical impedance spectroscopy (EIS) of the hydrogel electrolyte. The helical transition on gelatin molecules is strongly influenced by temperature. This means that the Gel-SO${}_{3/4}^{2-}$ gel has the potential to achieve remelting and recasting cycles. As shown in Figure 3d, the Gel-SO${}_{3/4}^{2-}$ gel of positive heptagonal shape can be remelted and recast into uniform pentagons. In addition, the Gel-SO${}_{3/4}^{2-}$ gel can be remelted and recast into various small car models many times without affecting its thermoelectric properties and transparency, as shown in Figure 3e. This means that the A-gel can be realized with recyclable value. In addition, benefiting from the remelting and recasting properties of Gel-SO${}_{3/4}^{2-}$ gel, the self-healing function of Gel-SO${}_{3/4}^{2-}$ gel can be simply implemented, as shown in Appendix A. Appendix A shows the cyclic voltammograms of the thermoelectric gel containing redox pair SO${}_{3/4}^{2-}$ at 263 K, 273 K and 293 K with different sweep second rates, respectively. The degree of reversibility of the redox pair reaction can be judged by the symmetry of the redox wave in the cyclic voltammogram. If the curve of the peak is symmetrical, the reaction is reversible, while an asymmetrical curve is irreversible. Appendix A observes that the CV curves of the redox reaction are symmetrical for 10 excellent reversibility scan cycles at the temperatures of 263 K, 273 K and 293 K, respectively. Therefore, the reactions can be considered reversible.

Furthermore, we measured the thermoelectric output on a self-made temperature gradient platform, as shown in Figure 4a. The Seebeck coefficient is calculated in terms of the equation, $S_{e}=-\frac{△U}{△T}\frac{△D}{△d}$. In this equation, Parr patches were used as the hot end and cold end, respectively, and carbon paper was used as the conductive electrode, where △D is the distance between the two ends of the gel sample (mm), △d is the distance between the two ends of the electrode (mm), △T is the temperature difference between the two ends of the gel (K) and △U is the gel thermal potential (mV). Normally, △D≈△d; thus, $S_{e}$ can be expressed as $S_{e}=-\frac{△D}{△d}$. A physical photograph of the Seebeck coefficient test is shown in Appendix A.

The highest-recorded Seebeck coefficient for the thermogalvanic gel is 1.62 mV·K${}^{-1}$. A negative Seebeck coefficient indicates an n-type thermogalvanic material, while a positive Seebeck coefficient indicates a p-type material. The Seebeck coefficient is determined by the entropy difference of the solventized structure and the concentration ratio difference of the redox components, as per thermodynamic theory. Appendix A provides a schematic illustration of the thermopower generation within the complex gel system, which encompasses Na${}^{+}$, Cl${}^{-}$, SO${}_{3}^{2-}$, SO${}_{4}^{2-}$, H${}_{2}$O and the gelatin cross-linking structure. This diagram helps visualize the thermoelectric power-generation process and the interactions among the different components within the gel. In Appendix A, when a temperature difference of 3 K is applied to both ends of the gel, the voltage reaches a maximum value of 4.86 mV in 30 s. When the heat source is withdrawn, the potential decreases with the temperature difference. Further, we tested the thermoelectric potential output signal at 0, 2, 4, 6, 8 and 10 K temperature differences (Figure 4b). It is observed that the thermoelectric potential increases linearly with the rise of temperature difference, indicating that the Gel-SO${}_{4}^{2-}$ gel has a good linear relationship of thermal-electrical conversion.

When one terminal of the gel thermocell is maintained at 273 K, the output power/current–voltage curves are measured at various temperature differences (△T). At temperatures above room temperature, the thermocell exhibits a short-circuit current of approximately 12 μA, an open-circuit voltage of around 4.8 mV and a maximum power output of about 14.4 μW at a △T of 3 K (Figure 4c,d). As the temperature difference (△T) increases to 20 K, the maximum power output increases to approximately 1.4 mW·m${}^{-2}$. To improve the long-term operating stability of the gel, the gel device is encapsulated with PU film, as shown in Appendix A. Based on the reliable temperature response exhibited by the gel, we utilized the Gel-SO${}_{3/4}^{2-}$ gel to monitor the temperature of both hot water and iced water, as shown in Appendix A. When the Gel-SO${}_{3/4}^{2-}$ gel was placed near the hot water cup, it produced an open-circuit voltage of approximately 50mV. Conversely, when the A-gel was placed near the cold water cup, it generated an open-circuit voltage of approximately −24 mV (Figure 4e,f). To further investigate the temperature-detection capabilities of the Gel-SO${}_{3/4}^{2-}$ gel for different food items, a selection of four representative foods was chosen for testing, as depicted in Appendix A. The purpose of this test was to evaluate the performance of the thermoelectric device in monitoring the temperatures of various food samples. The results of the test revealed intriguing findings. For rice samples with temperatures higher than room temperature (approximately 323 K as measured by the temperature tester), the thermoelectric device exhibited an output of approximately 52 mV of thermal potential. This observation clearly demonstrates the ability of the thermoelectric device to effectively monitor the temperatures of hot food items. Conversely, when subjected to foods with lower temperatures, such as ice cream, the Gel-SO${}_{3/4}^{2-}$ gel still yielded a significant thermal potential of around 38 mV. This indicates that the thermoelectric device is capable of accurately detecting and monitoring the temperature of cold food items as well. Similarly, for foods like coffee and apples at room temperature, the Gel-SO${}_{3/4}^{2-}$ gel displayed discernible monitoring abilities, as evidenced by the measured thermal potentials. These findings highlight the versatility and effectiveness of the A-gel as a temperature-sensing material for a wide range of food items. The ability of the thermoelectric device to accurately capture temperature variations in both hot and cold food samples opens up new possibilities for its application in the field of food safety and quality monitoring. These results clearly demonstrate the ability of the Gel-SO${}_{3/4}^{2-}$ gel to accurately detect and reflect temperature variations, providing a practical solution for temperature-monitoring applications. These significant voltage and current output signals form the Gel-SO${}_{3/4}^{2-}$ gel validate the efficacy of the designed thermogalvanic gel in achieving efficient thermal-electrical linear conversion under different conditions.

## 4. Conclusions

To summarize, this work presents the design of an adaptable and stretchable Gel-SO${}_{3/4}^{2-}$ thermogalvanic gel, which combines the use of Gel-SO${}_{3/4}^{2-}$ as a redox pair and gelatin as a polymer network. The resulting thermogalvanic gel exhibits exceptional properties that make it suitable for use as a wearable electronic device with direct skin contact. The gel offers a range of compelling features, including satisfactory thermogalvanic performance, non-toxicity, flexibility, stretchability and recyclability. The Gel-SO${}_{3/4}^{2-}$ thermogalvanic gel demonstrates excellent temperature response, enabling its application as a flexible electronic skin for self-driven sensing of micro- and nanoelectronic devices. By attaching gel patches to food, real-time monitoring and sensing capabilities can be provided to facilitate a variety of applications in the field of passive electronics. The gel’s adaptability and stretchability make it highly suitable for conformal adhesion to the human skin, ensuring a comfortable and seamless user experience. Its non-toxic nature ensures the safety of the user, eliminating potential risks associated with toxic materials. Moreover, the gel’s flexibility and stretchability allow for unrestricted movement, enabling it to accommodate various body movements without compromising its functionality. Additionally, the recyclability of the Gel-SO${}_{3/4}^{2-}$ thermogalvanic gel further contributes to its sustainability. The gel can be easily reused or recycled, minimizing waste and promoting environmental responsibility. Overall, the Gel-SO${}_{3/4}^{2-}$ thermogalvanic gel presents a promising solution for the development of wearable electronic devices that can be directly applied to the skin. Its outstanding characteristics, including thermogalvanic performance, non-toxicity, flexibility, stretchability and recyclability, make it an ideal candidate for various applications in the field of wearable electronics, particularly in the realm of self-driven sensing for micro- and nanoelectronic devices.

## Figures and Tables

**Figure 1 micromachines-14-01327-f001:**
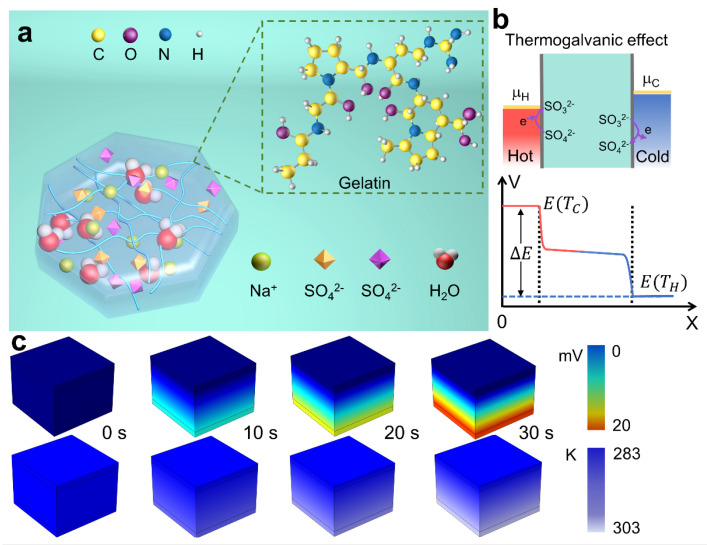
Design of the Gel-SO${}_{3/4}^{2-}$ gel. (**a**) The schematic diagram of multifunctional gel. (**b**) Electrochemical potential (μ) and the corresponding voltage (V) distribution in the thermogalvanic effect. (**c**) Heat transfer process and potential difference simulated via finite element method.

**Figure 2 micromachines-14-01327-f002:**
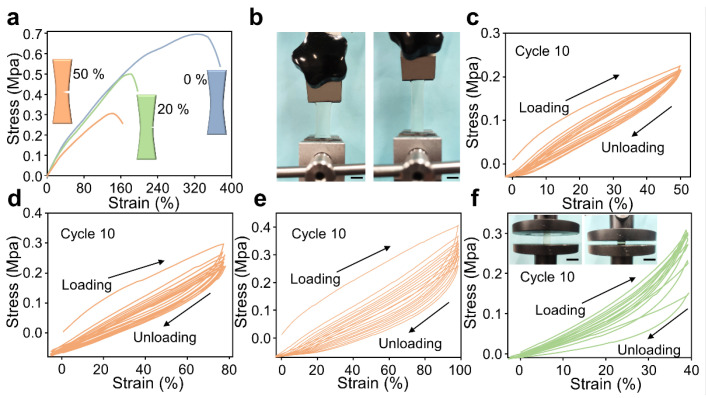
Mechanical properties of Gel-SO${}_{3/4}^{2-}$ gel. (**a**) Stress–strain curves for different notch sizes (T${}_{ave}$ = 298 K). (**b**) Physical photo of the tensile strength test. (**c**–**e**) Compression of the PG gel was performed at room temperature with 3 different degrees of repetitive tensile curves (T${}_{ave}$ = 298 K). (**f**) Cyclic loading–unloading curves of hydrogel at a compression strain of 40%; the insets show a photograph of the hydrogel under compression.

**Figure 3 micromachines-14-01327-f003:**
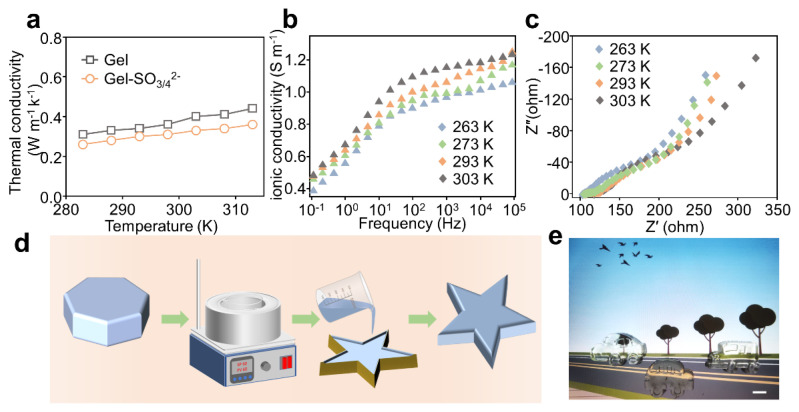
Thermal conductivity, electrical conductivity and recastability of Gel-SO${}_{3/4}^{2-}$ gel. (**a**) Thermal conductivity of Gel-SO${}_{3/4}^{2-}$ gels with and without SO${}_{3/4}^{2-}$. (**b**) Frequency dependence of Gel-SO${}_{3/4}^{2-}$ gel conductivity at different temperatures. (**c**) EIS spectra with different temperature. (**d**) Gel-SO${}_{3/4}^{2-}$ gel remelt, recast schematic. (**e**) Physical photograph of Gel-SO${}_{3/4}^{2-}$ gel being recast in the shape of small cars.

**Figure 4 micromachines-14-01327-f004:**
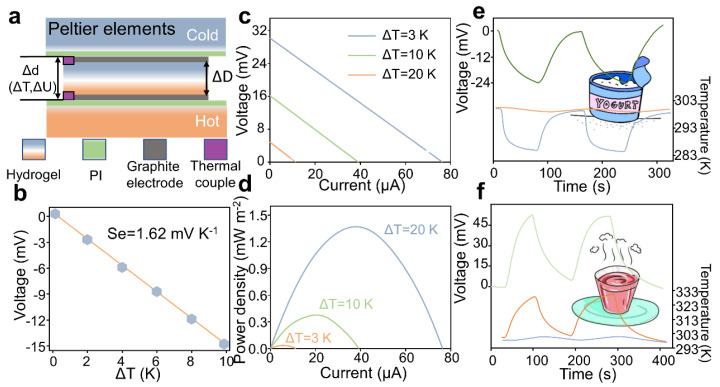
Thermoelectric properties of the thermogalvanic gel. (**a**) Schematic image of the platform for thermopower measurements. (**b**) Temperature–voltage curves under different temperature difference conditions. (**c**,**d**) Current–voltage curves and the corresponding power densities of the Gel-SO${}_{3/4}^{2-}$ gel thermogalvanic cell at different temperatures. The temperature at one end was fixed at T = 293 K and △T = $T_{h}-T_{c}$. (**e**,**f**) Gel-SO${}_{3/4}^{2-}$ gel is used to monitor the temperature of food; (**e**) cold food; (**f**) hot food.

## Data Availability

The authors confirm that the data supporting the findings of this study are available within the article and Appendix A.

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
