# Peer review of "Self-Powered, Non-Toxic, Recyclable Thermogalvanic Hydrogel Sensor for Temperature Monitoring of Edibles"

_micromachines, 2023, doi:10.3390/mi14071327_

Round 1

Reviewer 1 Report

Herein, the authors proposed a self-powered, non-toxic, recyclable temperature sensor based on thermogalvanic gel for edibles temperature monitoring. In my viewpoint, this paper may be acceptable if the following issues could be addressed.

1. The caption of Figure 3 seems a little chaotic. Some expressions are confused to readers. Please check and revise them.

2. In the Introduction section, the authors stated that "The rise of the wearable electronics market has driven the demand for self-powered energy sources and generated interest in low-grade waste heat collection ". More references should be cited.

3. Have the authors considered further packaging of the device to obtain long-term stability.

4. For food monitoring applications, why not do a monitoring comparison of several different foods?

Author Response

the responses of the comment's made by reviewer1 is listed in attchment

Reviewer 2 Report

In this study, the authors have successfully developed a novel Gel-SO2 3/4 − thermogalvanic gel that exhibits remarkable adaptability and stretchability. This gel system utilizes the redox pair SO2 3/4 − and gelatin, a polymer network, resulting in a unique material with a wide range of potential applications. A notable characteristic of this gel is its ability to be utilized as a wearable electronic device that directly interfaces with the skin. The study is intriguing and presents promising advancements in this field.

In the introduction, it would be beneficial for the authors to mention related thermogalvanic hydrogel gels, such as the paper Contraction waves in self-oscillating polymer gels or Periodical propagation of torsion in polymer gels, even though they involve toxic redox couples. Additionally, the authors mentioned that their chosen redox couple is non-toxic. It would be valuable if they could provide some biological data or references supporting this claim.

The mechanism of the self-powered system in their gel remains unclear. It would be helpful if the authors could provide a schematic diagram to illustrate the working principle. Additionally, it would be beneficial for them to explain the input energy source for the system.

In Figure 3b, the authors show the frequency-dependent ionic conductivity. It would be important for the authors to clarify the meaning of the frequency and how they determined the specific frequency used in their experiments.

To enhance the clarity for readers, it would be advantageous for the authors to present the experimental process and provide optical images of the gels used in their study.

Overall, this paper presents an interesting study with potential implications in the field.

Author Response

the responses of the comment's made by reviewer2 is listed in the attchment

Round 2

Reviewer 2 Report

The authors did a nice revision.